# Impact of an integrated intervention package during preconception, pregnancy, and early childhood on inflammation, IGF-1, IGFBP3 during first 6 months of life: Findings from the WINGS randomized controlled trial

Ranadip Chowdhury[1,2]*, Urmimala Maiti[1], Sarita Devi[3], Roshni M. Pasanna[3], Sunita Taneja[1], Partha P. Majumder[4], Tor A. Strand[5], R M Pandey[6], Anura V. Kurpad[3], Souvik Mukherjee[7,8], Nita Bhandari[1]

1 Society for Applied Studies, New Delhi, India, 2 DBT and Wellcome India Alliance Clinical and Public Health Fellow, Hyderabad, India, 3 Department of Physiology, St. John's Medical College, Bengaluru, India, 4 Human Genetics Unit, Indian Statistical Institute, Kolkata, West Bengal, India, 5 Department of Research, Innlandet Hospital Trust, Lillehammer, Norway, 6 Department of Biostatistics, All India Institute of Medical Sciences, New Delhi, India, 7 Human Microbiome Research Laboratory, Biotechnology Research Innovation Council-National Institute of Biomedical Genomics, Kalyani, West Bengal, India, 8 Regional Centre for Biotechnology, NCR Biotech Science Cluster, Faridabad, India

* ranadip.chowdhury@sas.org.in

## Abstract

Early-life interventions targeting maternal and child health, nutrition, and psychosocial care may influence infant growth and immune development by modulating systemic inflammation and growth factor pathways. However, causal evidence on comprehensive, integrated interventions initiated during early life remains limited. This study was nested in a factorial randomized controlled trial involving women aged 18–30 years. Participants were randomized to receive either a preconception intervention package or routine care until early childhood. The intervention included health care for growth-related conditions, nutrition, WASH (water, sanitation, and hygiene), and psychosocial care. The primary study demonstrated a substantial effect of the intervention on child growth and development. Blood samples from 381 infants (178 in intervention and 203 in control group) were analyzed at 3 and 6 months of age for inflammatory and growth-related biomarkers: C-reactive protein (CRP), alpha-1-acid glycoprotein (AGP), insulin-like growth factor 1 (IGF-1), and IGF binding protein 3 (IGFBP3). Generalized linear models were used to estimate mean differences and relative risks for the outcomes. The trial was registered at the Clinical Trials Registry–India: CTRI/2020/10/028770. Baseline maternal characteristics were similar between the two groups. At 3 and 6 months, the proportions of infants with CRP levels > 5 mg/L were similar in both groups. No significant differences were observed in AGP or IGF-1 concentrations at either time point. IGFBP3 was lower in the intervention group at 3 months [adjusted mean difference: [30.8(−55.3, −6.3)] ng/mL, but this effect was

**Data availability statement:** https://doi.org/10.5061/dryad.6t1g1jxbs.

**Funding:** The DBT (Department of Biotechnology, Government of India)/ Wellcome Trust India Alliance supported the study through a Clinical and Public Health Research Intermediate Fellowship grant awarded to Dr. Ranadip Chowdhury (Clinical and Public Health Research Intermediate Fellowship grant to RC) [grant IA/CPHI/20/1/505247]. URLs to sponsors' websites: https://www.indiaalliance.org/ The funders had no role in study design, data collection and analysis, decision to publish, or preparation of the manuscript.

**Competing interests:** The authors have declared that no competing interests exist.

not sustained at 6 months. An integrated intervention delivered from preconception through pregnancy and early childhood did not reduce systemic inflammation markers or produce sustained improvements in growth factor profiles during the first six months of life. These findings highlight the complexity of early-life inflammatory processes and underscore the need for further research on long-term effects of early-life interventions in low-resource settings.

## Introduction

The continuum from preconception through pregnancy and early childhood constitutes a critical window for influencing growth, development, and long-term health outcomes. Substantial evidence suggests that environmental exposures and targeted health and nutrition interventions during these critical periods can have lasting effects on metabolic programming, immune system maturation, and developmental trajectories [1–3]. Of particular interest are the complex interplay between systemic inflammation (indicated by raised C-reactive protein or alpha-1-acid glycoprotein levels) and key growth mediators such as insulin-like growth factor 1 (IGF-1) and its principal binding protein, IGFBP3, given their integrated roles in somatic growth, neurodevelopment, immune regulation, and susceptibility to disease [4,5]. Persistent low-grade inflammation in early life has been linked to suboptimal growth and an increased risk of non-communicable diseases (NCDs). In contrast, a diverse and resilient gut microbiome, together with physiologically appropriate concentrations of IGF-1 and IGFBP3, is essential for supporting normal linear growth and optimal neurodevelopment [6]. Interventions addressing maternal and child nutrition, infection control, and psychosocial stimulation have demonstrated the potential to modulate these biological pathways and improve child health outcomes positively [7]. There is a paucity of research evaluating the effects of comprehensive, integrated intervention strategies initiated preconceptionally and sustained through pregnancy and early childhood on inflammatory profiles and growth factor dynamics in infants. To address this gap, we recently conducted the Women and Infants Integrated Growth Study (WINGS), an individually randomized factorial trial designed to assess the impact of a multidomain intervention package including health, nutrition, psychosocial care and support, and water, sanitation, and hygiene (WASH) delivered from preconception through early childhood [8]. The primary outcomes demonstrated that this integrated intervention significantly reduced the incidence of low birth weight (LBW), small for gestational age (SGA), and stunting at 24 months compared to standard care.

Building on this evidence, we designed the present study to evaluate the effects of the WINGS intervention on inflammatory markers and IGF-1/IGFBP3 concentrations during the first six months of life. By concurrently targeting maternal and infant health, nutrition, and environmental exposures during critical developmental windows, the intervention was expected to promote a healthier gut microbial milieu, attenuate systemic inflammation, and support more favourable growth factor profiles in infancy [9].

We hypothesized that this integrated package would result in reduced levels of inflammatory markers and improved levels of IGF-1 and IGFBP3 in the intervention group compared to the routine care group. In doing so, this study aims not only to assess the clinical impact of WINGS but also to elucidate how multi-domain early-life interventions in low-resource settings modulate inflammation and growth factor pathways, thereby clarifying the biological mechanisms that link such interventions to improved growth and developmental outcomes. We chose 3 and 6 months as measurement time points to capture early postnatal windows as infant immune and endocrine systems are rapidly maturing and assessment at these intermediate endpoints help elucidate early biological pathways potentially underlying the later anthropometric effects observed at 24 months.

## Materials and methods

### Ethics statement

Ethical clearance for this study was granted by the Ethics Review Committee of the Society for Applied Studies in New Delhi, India (Ref no. SAS/ERC/WINGS-Sub Study/2020). The study complied with ethical standard outlined in Declaration of Helsinki. This study was registered with the Clinical Trial Registry of India under the identifier CTRI/2020/10/028770. Written informed consent was secured primarily from mothers before their inclusion in the study.

### Primary study design

This was a factorial randomized controlled trial (RCT) conducted in low- and middle-income neighbourhoods of Delhi, India. Eligible women of reproductive age (18–30 years), identified through door-to-door survey were enrolled and randomized to receive either the preconception intervention package or routine care. They were followed for 18 months. If they were identified as pregnant in this period (ultrasonographic confirmation of pregnancy), they were then randomized either to receive the pregnancy and early childhood intervention package or routine care. All pregnant women were followed until delivery, and their children until they reached 24 months of age. Data of those children were included in analysis who reached 24 months of age by 30th June 2021. Women were randomised using permuted blocks and stratified by maternal height (<150 cm and>= 150 cm). Group allocation (1:1) was through a web based system. The primary study has been detailed elsewhere by Taneja et al. [8]. This two-step randomization resulted in four groups: preconception and pregnancy interventions (A), preconception interventions only (B), pregnancy and early childhood interventions only (C), and no preconception interventions, with routine pregnancy and early childhood care (D). The interventions, under the four domains of health, nutrition, WASH, and psychosocial care and support, are briefly described below.

In the pre-conception period, women were screened and treated for medical conditions, including anemia. All received one tablet of iron and folic acid (IFA) daily and weekly multiple micronutrients (MMN) supplements to meet ½ to ¾ of the recommended daily allowance (RDA). They also received an albendazole tablet twice a year. All women were counseled on adequate diets, positive thinking, problem-solving skills, and menstrual and hand hygiene. We could not mask participants and teams because of the nature of the interventions. Outcomes were assessed by an independent team not involved in delivering interventions or aware of the group allocation before measurements. (Full proposal attached as S1 Text).

### Intervention group

In the pregnancy and early childhood interventions, women received monthly antenatal care, screening, and treatment for medical conditions, including anemia. Women received daily IFA, MMN (~1 RDA), calcium, and vitamin D throughout pregnancy. Albendazole was given once during pregnancy. Women with BMI < 25 kg/m² in the second and third trimesters received food supplements. Women with inadequate weight gain (IWG) were identified based on the Institute of Medicine guidelines for gestational weight gain (GWG), and nutritional counseling and extra food supplements were provided [8,10].

Water filters, water storage bottles, hand washing stations, soap, and disinfectants were provided to all families, in addition to counseling on sanitation and hygiene. Women received counselling on exclusive breastfeeding during the first 6 months of the infant's age. Additional visits were made for babies born preterm, LBW, and for mothers with breastfeeding problems. Nutritional supplementation included vitamin D for all infants, iron for very low and low birth weight infants, and daily snacks and supplements for mothers. Complementary food supplements (milk-cereal mix) were started at six months and continued until 24 months. Mothers were taught age-specific child play, responsive care, and stimulation activities. Weights were measured during home visits, and children with inadequate weight gain were referred to lactation counselors and pediatricians.

### Control group

Women in the control group were advised to seek care from government sources (free of cost) to access family planning services and weekly iron folic acid supplementation. They were advised to register for antenatal care at a government or private facility, and have at least four antenatal care checkups, consume iron folic acid, calcium, vitamin D daily throughout pregnancy, access supplementary foods through the Integrated Child Development Services (ICDS) scheme and plan to deliver in health facilities. Women were advised to go for a postnatal health check-up and to consume iron, folic acid, calcium, vitamin D, and supplementary foods daily through the ICDS scheme. Mothers were advised to breastfeed their babies exclusively for the first six months and continue breastfeeding for at least two years. They were also encouraged to arrange home visits by the community health workers in the first 42 days of life.

### Participants of the current study

Between 1st December 2020 and 30th December 2021, 381 mother-infant dyads were selected consecutively when the infants completed 6 months of age from Group A and D—as defined in the primary WINGS study. Randomization was conducted by an independent statistician, who provided a list to ensure unbiased group allocation. All outcome assessments were conducted by a separate team that was not involved in the intervention delivery and was blinded to group assignments before the measurement process.

### Sample size

Few intervention studies have systematically examined the infant blood markers of inflammation and growth during the first six months of life. In the WINGS study, we hypothesized a 0.2 standard deviation (SD) increase in Length-for-Age Z-score (LAZ) and a 25% relative reduction in the incidence of stunting at 24 months in the group receiving interventions during preconception, pregnancy, and early childhood (Group A), compared to the group receiving routine care (Group D) [5]. For the present study, we hypothesized that the intervention package would result in increased IGF-1 levels and reduced inflammatory markers, such as CRP, with an anticipated effect size equal to or greater than that observed in the WINGS study. Assuming a corresponding 0.3 SD decrease of CRP in the intervention group with the power at 80% and 95% confidence level, we calculated that a total of 352 infant blood samples would be required —176 in the intervention group and 176 in the routine care group.

### Sample collection and laboratory analysis

We collected 3 mL of blood from infants at 3 and 6 months of age. The samples were transported from the field to the "Clinical and Research laboratory" in SAS at 4°C on ice packs and centrifuged to extract serum. The serum was then stored at -80°C for analysis of IGF-1, IGBP3, CRP, and AGP.

Serum IGF-1 and IGFBP3 were analyzed by electrochemiluminescence immunoassay on the Cobas e 601 analyzer (Roche Diagnostics, Mannheim, Germany) using the Elecsys IGF-1 kit (Roche Diagnostics, Ref No. 07475896190), and

IGFBP-3 kit (Roche Diagnostics, Ref No. 07574690190), according to the manufacturer's instructions. Calibration and internal quality controls were performed using Roche-provided calibrators and control materials. Results were expressed in nanograms per milliliter (ng/mL). Intra and Inter assay CVs were <3 and <4% respectively.

Serum CRP and AGP were measured by immunoturbidometric based assay on the Cobas c 501 analyzer (Roche Diagnostics, Mannheim, Germany). All reagents, calibrators, and controls were obtained from Roche Diagnostics, and the assay was performed according to the manufacturer's guidelines. Quality control procedures were performed using standard control materials supplied by the manufacturer. Results were expressed in milligrams per liter (mg/L) and grams per liter (g/L) for CRP and AGP, respectively. All samples were analyzed in a single batch to reduce inter-assay variability. Intra- and inter-assay CVs were <2% and <4%, respectively.

### Analysis plan

Baseline characteristics of the preconception, pregnancy, and early childhood group (Group A) and the routine care group (Group D) were compared using means and standard deviations (SD), median and interquartile range (IQR) for continuous variables, and proportions for categorical variables. All analyses followed an intention-to-treat approach and were conducted using STATA version 16. Generalized linear models (GLMs) with a Gaussian family and identity link function were used to estimate mean differences in CRP, AGP, IGF-1, and IGFBP3 concentrations. To estimate risk ratios for inflammatory status between infants in the intervention and routine care groups, GLMs with a binomial family and log link function were employed. Final models were adjusted for place of birth.

## Results

### Baseline characteristics

Between December 2020 and December 2021, 381 infants (178 in the preconception, pregnancy, and early childhood intervention group, and 203 in the routine care group) were recruited for this study (Fig 1). The baseline characteristics of the women were similar, except for the proportion of underweight women, the number of families possessing a below the poverty line (BPL) card, and the place of birth (Table 1, Fig 2).

Table 2 compares infant inflammatory and growth-related biomarker levels during the first 6 months of life between the Preconception, Pregnancy, and Early Childhood group and the Routine Care group. CRP, AGP, and IGF-1 levels were similar between the groups at both 3 and 6 months. IGFBP3 levels at 3 months were lower in the Preconception, pregnancy, and early childhood group compared to the Routine care group (adjusted mean difference: -32.49 ng/mL; 95% CI: -56.79, -8.19), but this difference was not sustained at 6 months.

Table 3 compares the prevalence of elevated inflammatory markers in infants during the first six months of life between the Preconception, Pregnancy, and Early Childhood group and the Routine Care group. The elevated inflammatory status was similar at 3 and 6 months of life in both the Preconception, Pregnancy, and Early Childhood group and the Routine Care group. The intervention group had a lower proportion of infants with CRP > 5 mg/L compared to the routine care group at 3 months. However, this effect was attenuated by adjusting for potential confounders. [adjusted Relative risk (RR) = 0.49; 95% CI: 0.22, 1.09].

## Discussion

In this randomized controlled trial, we evaluated the impact of an integrated intervention package delivered from preconception through pregnancy and early childhood on infant inflammatory markers and growth-related hormones during the first six months of life.

Evidence from previous studies indicates that healthy children exhibit lower concentrations of CRP and AGP than malnourished children, and higher AGP levels have been associated with reduced height-for-age Z-scores [11,12]. Additionally, maternal micronutrient supplementation has been shown to alter maternal inflammatory biomarkers, which may subsequently

**Figure 1. Screening, enrollment, randomization and follow up**

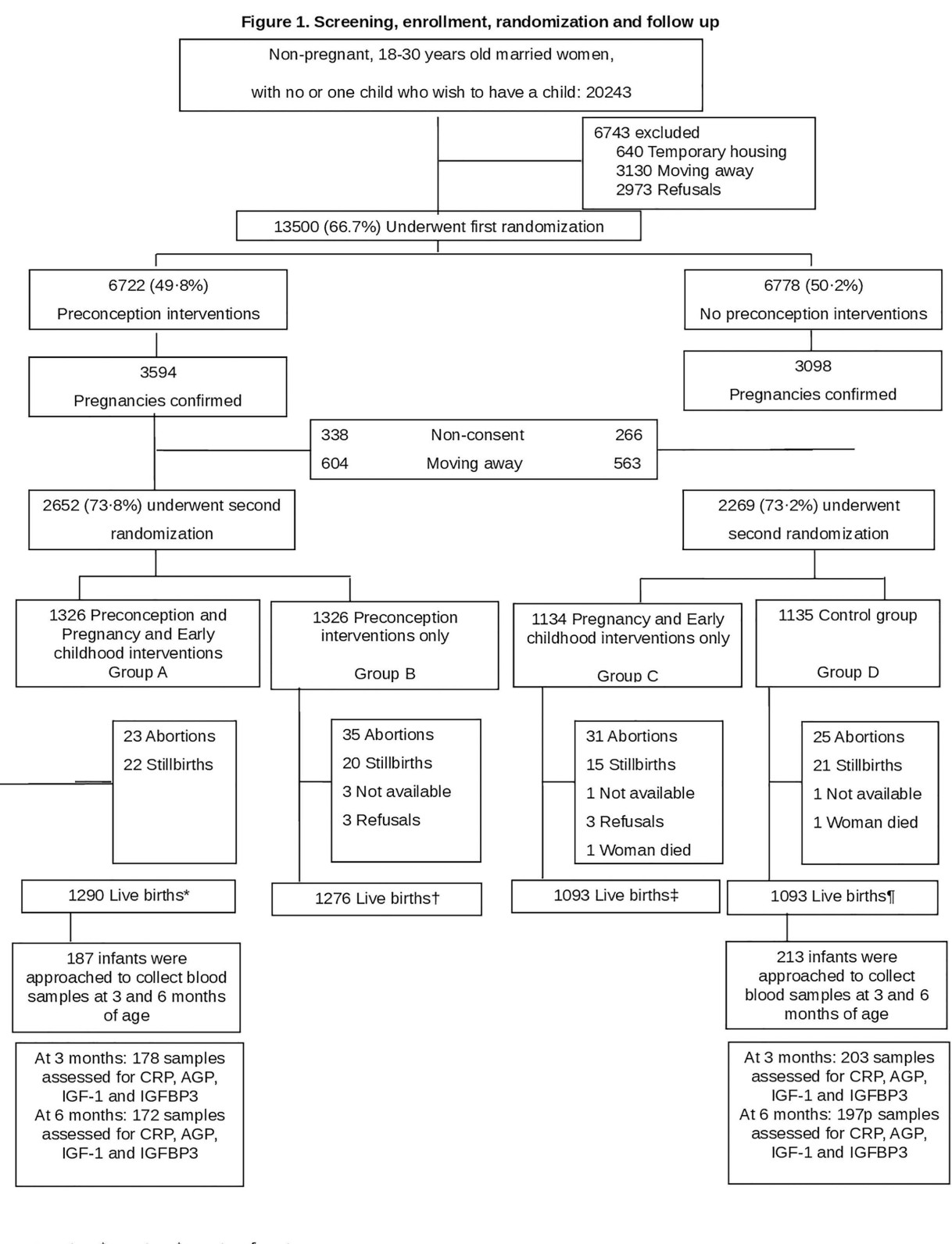

*9 twins, †11 twins, ‡10 twins, ¶6 twins

**Fig 1. Screening, enrolment, randomization and follow up.**

**Table 1. Baseline characteristics of mother at recruitment.**

| | Preconception, pregnancy and early childhood (Group A) (n=178) | Routine care (Group D) (n=203) |
|---|---|---|
| Woman age (years) | 24.02 (2.99) | 23.99 (3.09) |
| Woman height (cm) | 151.99 (5.27) | 152.40 (5.75) |
| BMI category (kg/m²); n (%) | | |
| >=25 | 44 (24.72) | 48 (23.64) |
| 18.5 - 24.99 | 107 (60.11) | 118 (58.13) |
| <18.5 | 27 (15.17) | 37 (18.23) |
| Joint/ Extended family; n (%) | 109 (61.24) | 127 (62.56) |
| Women schooling >12 yrs; n (%) | 103 (57.87) | 107 (52.71) |
| Homemaker; n (%) | 174 (97.75) | 195 (96.06) |
| Family has BPL card; n (%) | 6 (3.37) | 16 (7.88) |
| Family covered by health insurance scheme; n (%) | 11 (6.18) | 20 (9.85) |
| Place of delivery; n (%) | | |
| Large hospital | 124 (69.66) | 68 (33.50) |
| Middle level hospital/ small hospital/ birthing centre | 50 (28.09) | 116 (57.14) |
| Home | 4 (2.25) | 19 (9.36) |

*2 twins 1 in Preconception, pregnancy and early childhood group and 1 in Routine care group; n (%) unless specified otherwise; BPL= Below Poverty Line; All values expressed as Mean (SD) unless specified otherwise.

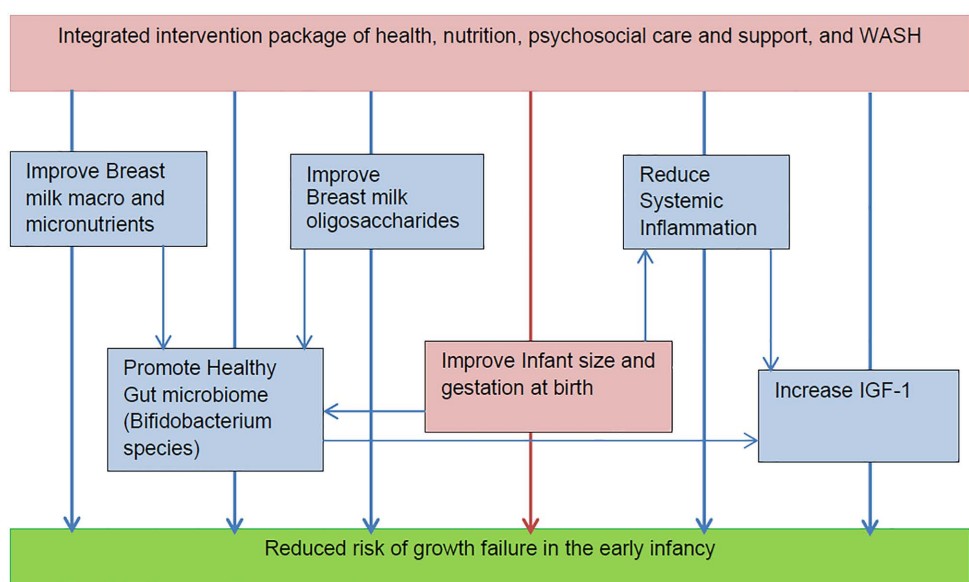

**Fig 2. Conceptual framework: Postulated pathways of intervention effect to reduce growth faltering first 6 months of age.**

**Table 2. Infant inflammatory growth biomarkers levels during the first 6 months of life.**

| Inflammatory markers and growth-related hormones | Preconception, pregnancy and early childhood (Group A) (n=178) | Routine care (Group D) (n=203) | Unadjusted mean difference | Adjusted mean difference |
|---|---|---|---|---|
| CRP at 3 months (mg/L) Mean (SD) | 2.20 (7.64) | 3.16 (9.78) | -0.96 (-2.74, 0.82) | -0.65 (-2.55, 1.25) |
| Median (IQR) | 0.59 (0.58, 0.71) | 0.59 (0.58, 1.06) | | |
| CRP at 6 months (mg/L) Mean (SD) | 2.42 (7.62) | 1.66 (4.13) | 0.76 (-0.47, 1.99) | 1.03 (-0.29, 2.36) |
| Median (IQR) | 0.59 (0.58, 0.66) | 0.59 (0.58, 0.59) | | |
| AGP at 3 months (g/L) Mean (SD) | 0.55 (0.24) | 0.57 (0.29) | -0.02 (-0.07, 0.03) | -0.02 (-0.08, 0.03) |
| Median (IQR) | 0.5 (0.36, 0.64) | | 0.5 (0.35, 0.7) | |
| AGP at 6 months (g/L) Mean (SD) | 0.58 (0.38) | 0.56 (0.29) | 0.02 (-0.05, 0.08) | 0.01 (-0.06, 0.08) |
| Median (IQR) | 0.46 (0.37, 0.66) | 0.49 (0.38, 0.65) | | |
| IGF-1 at 3 months (ng/mL) Mean (SD) | 9.84 (4.90) | 10.03 (5.29) | -0.19 (-1.22, 0.84) | -0.36 (-1.46, 0.75) |
| Median (IQR) | 6.99 (6.98, 11.21) | 6.99 (6.98, 11.7) | | |
| IGF-1 at 6 months (ng/mL) Mean (SD) | 8.94 (4.40) | 9.26 (4.48) | -0.32 (-1.23, 0.59) | -0.30 (-1.29, 0.67) |
| Median (IQR) | 6.99 (6.98, 8.69) | 6.99 (6.98, 8.83) | | |
| IGFBP3 at 3 months (ng/mL) Mean (SD) | 181.20 (114.70) | 210.04 (109.95) | -28.84 (-51.60, -6.09) | -32.49 (-56.79, -8.19) |
| Median (IQR) | 172 (78, 273) | 222 (112, 303) | | |
| IGFBP3 at 6 months (ng/mL) Mean (SD) | 1523.42 (497.45) | 1496.46 (475.81) | 26.95 (-72.72, 126.63) | -5.16 (-112.28, 101.95) |
| Median (IQR) | 1504 (1259, 1764) | 1499.5 (1201.5, 1800.5) | | |

Note: Values are means±SDs (n) or mean differences (95% CI); Adjustment was done for place of delivery; IQR: Interquartile range.

**Table 3. Infant inflammatory status during first 6 months of life.**

| Inflammatory markers | Preconception, pregnancy and early childhood (Group A) (n=178) | Routine care (Group D) (n=203) | Unadjusted RR (95% CI) | Adjusted RR (95% CI) |
|---|---|---|---|---|
| CRP>5 (mg/L) at 3 months | 9 (5.06) | 22 (10.84) | 0.47 (0.22, 0.98) | 0.49 (0.22, 1.09) |
| CRP>5 (mg/L) at 6 months | 19 (10.67) | 18 (8.87) | 1.20 (0.65, 2.22) | 1.18 (0.61, 2.28) |
| AGP>1 (g/L) at 3 months | 11 (6.18) | 13 (6.40) | 0.96 (0.44, 2.10) | 0.92 (0.40, 2.14) |
| AGP>1 (g/L) at 6 months | 21 (11.80) | 20 (9.85) | 1.20 (0.67, 2.22) | 1.1 (0.60, 2.10) |

Note: Adjusted for place of delivery, RR: Relative risk.

affect metabolic outcomes in the offspring [11,12]. Building on this evidence our study found that CRP, AGP, and IGF-1 levels were similar between the intervention and routine care groups at both 3 and 6 months. IGFBP3 levels at 3 months were lower in the intervention group compared to the routine care group, but this difference was not sustained at 6 months.

These results suggest that, despite the comprehensive nature of the intervention, which included nutritional supplementation, infection control, WASH improvements, and psychosocial support, there was no consistent or sustained effect on systemic inflammation or growth factor profiles in early infancy [13,14]. Several factors may explain the lack of differences observed in our study. First, the prevalence of raised inflammatory markers in this population was relatively low, which may have limited the potential for further reduction in CRP and AGP even with an intensive intervention. Second, the IGF-1/IGFBP-3 axis in early infancy is strongly driven by age-related maturation, genetic programming and basic energy/protein sufficiency. Modest short-term improvements in nutrition, WASH, infection control, and psychosocial stimulation are unlikely to override these intrinsic trajectories within 3–6 months, attenuating the impact of the comprehensive intervention package.

The transient reduction in IGFBP3 at 3 months in the intervention group without a corresponding rise in IGF-1 likely reflects normal developmental variation or a small, short-lived change insufficient to alter total IGF-1 or tissue-level IGF-1 activity. Previous studies have demonstrated a negative correlation between inflammatory markers and growth-related hormones, i.e., a rise in CRP and AGP corresponded with a decrease in IGF-1 and IGFBP3 levels [4]. IGFBP3 is the primary binding protein for IGF-1, playing a crucial role in regulating IGF-1 bioavailability and activity. While lower IGFBP3 could theoretically enhance IGF-1 action, we did not observe corresponding increases in IGF-1 concentrations at this early time point. This case control study in Zimbabwe found that IGF-1 and IGFBP-3 decreased with time from 3 months of age among both stunted and non-stunted children [4]. A prospective cohort study on healthy infants examining the effect of nutritional differences on growth factor levels found that dairy-based protein intake was associated with increased levels of both IGF-1 and IGFBP-3 between 9 and 12 months of age, compared to non-dairy animal source protein intake [15]. The biological significance of this isolated finding in our study is, therefore unclear.

Given the complexity of our intervention, the observed effects are likely due to the combined influence of multiple components, making it difficult to attribute changes to any single factor. Moreover, early-life inflammation is shaped by a complex interplay of perinatal factors, infections, gut health, microbiota, and oxidative stress, so systemic markers such as CRP and AGP may miss subtle or localized benefits of the intervention. The timing and duration of the intervention, while spanning critical developmental windows, may not have been sufficient to elicit measurable changes in systemic biomarkers within the first six months of life. Any effects of improved nutrition and environment on growth and endocrine function may emerge only later in infancy or childhood, and our intervention may have enhanced overall adequacy rather than the specific nutrient profiles (e.g., particular protein sources or micronutrients) known to more strongly modulate IGF-1 and IGFBP-3. Conclusions based solely on data from 3 and 6 months may be inappropriate. It is also possible that the benefits of the intervention on inflammation and growth factors may emerge later in childhood, as has been observed in some longitudinal studies [16]. However, that again will be subject to varying degrees of confounding factors over a long time period that can alter the levels of the inflammatory markers [17]. Further research is needed to translate the data on IGF-1 and IGFBP-3 into meaningful diagnostic outcomes, considering the influence of age, gender, and genetic factors, among other determinants.

Our study has several strengths, including its randomized design, rigorous implementation of a multidomain intervention, and careful adjustment for potential confounders such as place of delivery, socioeconomic status, and maternal BMI. The use of standardized laboratory methods and blinded outcome assessment further enhances the validity of our findings. However, some limitations should be acknowledged. The sample size, while adequate for detecting moderate effect sizes, may have been insufficient to identify smaller but clinically meaningful differences. Additionally, the study was conducted in a single urban setting in India, which may limit generalizability to other populations. Replicating the study in diverse socioeconomic and cultural settings would strengthen the evidence base and enable more conclusive interpretations [18].

## Conclusion

While early evidence points to the therapeutic effect of early nutritional and health interventions, this is one of the few prospective studies that have tried to evaluate the effect of the interventions on inflammatory processes. Our findings suggest

that an integrated intervention package delivered from preconception through early childhood do not reduce systemic inflammation or improve growth factor profiles during the first six months of life. These results suggest that additional or alternative strategies may be necessary to achieve meaningful improvements in these biomarkers, as well as the consideration of other biomarkers that reflect inflammatory responses. **These findings add to the understanding of biological mechanisms underlying integrated early-life interventions in LMICs.**

## Supporting information

**S1 Text. Study Proposal: Integrated Intervention Effects on Infant Inflammation and IGF Axis.** (PDF)

**S1 Consort Checklist. Hopewell S, Chan AW, Collins GS, Hróbjartsson A, Moher D, Schulz KF, et al.** CONSORT 2025 Statement: updated guideline for reporting randomised trials. BMJ. 2025; 388:e081123. https://dx.doi.org/10.1136/bmj-2024-081123. (DOCX)

## Acknowledgments

We deeply acknowledge the contribution of the participants and their families and are thankful to the community leaders for their cooperation.

## Author contributions

**Conceptualization:** Ranadip Chowdhury.

**Data curation:** Ranadip Chowdhury.

**Formal analysis:** Ranadip Chowdhury, Urmimala Maiti, R M Pandey.

**Funding acquisition:** Ranadip Chowdhury.

**Investigation:** Ranadip Chowdhury, Sarita Devi, Roshni M. Pasanna.

**Methodology:** Ranadip Chowdhury, R M Pandey.

**Project administration:** Ranadip Chowdhury, Sarita Devi, Roshni M. Pasanna.

**Resources:** Ranadip Chowdhury.

**Software:** Ranadip Chowdhury.

**Supervision:** Ranadip Chowdhury, Sarita Devi, Roshni M. Pasanna.

**Validation:** Ranadip Chowdhury.

**Visualization:** Ranadip Chowdhury.

**Writing – original draft:** Ranadip Chowdhury, Urmimala Maiti.

**Writing – review & editing:** Ranadip Chowdhury, Sunita Taneja, Partha P Majumder, Tor A. Strand, Anura V Kurpad, Souvik Mukherjee, Nita Bhandari.

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
