## [Decision Letter · Decision Letter 0]

30 Oct 2025

PGPH-D-25-02277

Title - Impact of an Integrated Intervention Package During Preconception, Pregnancy, and Early Childhood on inflammation, IGF-1, IGFBP3 during first 6 months of life: Findings from the WINGS Randomized Controlled Trial

Dear Dr. Chowdhury,

Thank you for submitting your manuscript to PLOS Global Public Health. After careful consideration, we feel that it has merit but does not fully meet PLOS Global Public Health’s publication criteria as it currently stands. Therefore, we invite you to submit a revised version of the manuscript that addresses the points raised during the review process.

We look forward to receiving your revised manuscript.

Kind regards,

Rahul Gajbhiye, MBBS PhD

Academic Editor

Journal Requirements:

i. Please clarify all sources of financial support for your study. List the grants, grant numbers, and organizations that funded your study, including funding received from your institution. Please note that suppliers of material support, including research materials, should be recognized in the Acknowledgements section rather than in the Financial Disclosure. 

ii. State the initials, alongside each funding source, of each author to receive each grant. For example: "This work was supported by the National Institutes of Health (####### to AM; ###### to CJ) and the National Science Foundation (###### to AM)."

iii. State what role the funders took in the study. If the funders had no role in your study, please state: “The funders had no role in study design, data collection and analysis, decision to publish, or preparation of the manuscript.”

2. Please ensure that your Ethics Statement is available in its entirety at the beginning of your Methods section, under a subheading 'Ethics Statement'. It must include:

1) The name(s) of the Institutional Review Board(s) or Ethics Committee(s)

2) The approval number(s), or a statement that approval was granted by the named board(s) 

3) (for human participants/donors) - A statement that formal consent was obtained (must state whether verbal/written) OR the reason consent was not obtained (e.g. anonymity). 

Additional Editor Comments (if provided):

Reviewers' comments:

Reviewer's Responses to Questions

**Comments to the Author**

1. Does this manuscript meet PLOS Global Public Health’s publication criteria?

Reviewer #1: Partly

Reviewer #2: Yes

Reviewer #3: Yes

2. Has the statistical analysis been performed appropriately and rigorously?

Reviewer #1: No

Reviewer #2: Yes

Reviewer #3: Yes

3. Have the authors made all data underlying the findings in their manuscript fully available (please refer to the Data Availability Statement at the start of the manuscript PDF file)?

Reviewer #1: Yes

Reviewer #2: Yes

Reviewer #3: Yes

4. Is the manuscript presented in an intelligible fashion and written in standard English?

Reviewer #1: Yes

Reviewer #2: Yes

Reviewer #3: Yes

Reviewer #1: This sub study was nested in a factorial randomized controlled trial (RCT) in women aged 18–30 years. Participants included in this study were randomized to receive either a preconception intervention package or routine care until early childhood. The design strategy involved a reasonable sample size justification to show superiority. The sample needed for the study objectives was well justified with power considerations. However, the investigators do note that the sample size, while adequate for detecting moderate effect sizes, may have been insufficient to identify smaller but clinically meaningful differences. The descriptives are informative as seen in Tables 1 and 2.

1. Please define IQR in the footnote of Table 2 or put a descriptive section in the ‘Analysis Plan’ paragraph.

Generalized linear models (GLMs) with a Gaussian family and identity link function were used to estimate mean differences in CRP, AGP, IGF-1, and IGFBP3 concentrations. To estimate risk ratios for inflammatory status between infants in the intervention and routine care groups, GLMs with a binomial family and log link function were employed. Final models were adjusted for place of birth. There are several considerations needing clarification.

There are four endpoints. Therefore,

2. Some consideration of multiple comparison p-value adjustment should have been discussed.

Also, with respect to model content,

3. Exactly how was adjustment by birthplace incorporated into the models?

The overall conclusions follow from the analyses performed and results seen in Table 3. The strengths and limitations are reasonably described in the ‘Discussion’ section. As an added point, however,

4.There is a gap between the manuscript text and the supplement supporting information proposal Version 2.0. Was there any attempt to explore the mediation analysis discussed in that proposal?

Reviewer #2: 1. Overall Assessment

This study reports a well-designed randomized controlled trial. It investigates the impact of an integrated intervention on infant biomarkers related to inflammation and growth like CRP, AGP, IGF-1, IGFBP3. The research addresses a significant question in maternal and child health.

However, the discussion sections can be improved with detailed explanation on biological plausibility. Also, the implications of this study can be broadly elaborated.

2. Originality and Relevance

The research topic appears to be original and highly relevant. The novelty in this study is integrated interventions across different stages right from preconception to 2 years of early child development. The intervention is policy-relevant and aligns well as per Goal-2 and Goal-4 of SDG-2030. The concept is innovative and similar integrated frameworks are reported in the literature.

The specific distinct approach of this study needs to be articulated.

3. Scientific Rigor and Methodology

This randomized controlled design follows standard protocols and manuscript is well-aligned as per CONSORT guidelines.

Please elaborate on randomization process, blinding, and control of confounders. The sample size calculations appear to be powered for anthropometric assessments. For biomarker outcomes, sample size calculations need to be refined/justified.

4. Results and Interpretation

The results of this study report no significant differences in biomarkers between intervention and control groups.

The null findings can be discussed with possible biological explanations like timing of assessment, nutritional variability, breastfeeding. Subgroup analysis by maternal or infant characteristics can be helpful.

5. Discussion and Implications

There is a scope to elaborate the discussion section by linking the pathways of maternal interventions with infant biomarker responses. Implications of this study for public health, including integration into maternal and child health programs, can be discussed highlighting the need for long-term follow-up.

6. Presentation and Clarity

The manuscript is well-written and well-organized as per required guidelines. However, most of the references are quite older and references from 2022 onwards are missing.

More recent Citations can be included from year 2023-2025.

7. Ethical and Data Considerations

All the ethical procedures are described clearly including IEC and CTRI. Data availability through Open Access links is provided.

8. Conclusion and Recommendation

This well-executed trial can be good evidence for understanding the biological outcomes of integrated maternal-child interventions.

Recommendation: Minor Revision.

Reviewer #3: This study is a secondary analysis of the WINGS factorial randomized controlled trial evaluating the effects of a multidomain, integrated intervention delivered from preconception through early childhood on infant biomarkers of inflammation and growth (CRP, AGP, IGF-1, IGFBP3) at 3 and 6 months of age. This study links the integrated intervention to specific changes in inflammatory and growth-related biomarkers like CRP, AGP, IGF-1 and IGFBP3. The study addressed the biologically relevant and policy-important question related to early-life interventions in low-resource settings

The findings indicate no significant differences in these biomarkers between the intervention and control groups, except for a transient decrease in IGFBP3 at 3 months, which was not sustained at 6 months. The authors conclude that while the intervention improved growth outcomes in the parent trial, it did not significantly influence early-life inflammation or IGF axis biomarkers.

The manuscript is well-written, clearly articulated and follows the required CONSORT Guidelines.

Major Comments

1. Rationale and Framing

• Biological rationale connecting integrated maternal–child interventions (nutrition, WASH, psychosocial care) with the specific biomarkers studied (CRP, AGP, IGF-1, IGFBP3), needs clarity

• Clarify why these markers and 3- and 6-month time points were selected, especially since primary growth outcomes were reported at 24 months in the main WINGS paper.

• A concise conceptual model or figure showing hypothesized pathways could help readers follow the mechanistic logic.

2. Study Power and Sample

• The power calculation is based on CRP only. Please justify the adequacy of the sample size for detecting meaningful differences in IGF-1 and IGFBP3, given their biological variability in infancy.

• Power calculations are based on LAZ outcomes from the primary WINGS study rather than biomarker data. This needs justification.

3. Statistical Analysis and results

• Tables 2 and 3 could be simplified to highlight group comparisons more effectively.

• Adjustment only for the place of delivery seems limited.

• The author may consider other covariates, such as mothers’ BMI, socioeconomic indicators, or exposure to infections, in the analysis. In case they are intentionally excluded from the analysis, explain their exclusion.

• It would be useful to include effect size interpretation (e.g., percentage change or standardized mean difference) to better convey the biological relevance of null findings.

4. Interpretation of Findings

• However, cautious interpretation of the null findings is needed. Aspects such as biological plausibility, contextual limitations, and future implications for longitudinal research require further elaboration.

• The discussion acknowledges the absence of significant effects, but can be deepened if the authors discuss the following issues

o Address low baseline inflammation as a potential ceiling effect.

o Note that intervention effects might appear later in life (after 6 months).

o Acknowledge that non-inflammatory mechanisms (caregiving, infection prevention, psychosocial stimulation) might explain the positive growth outcomes in the primary trial.

• Expand the comparison with similar trials—such as SHINE (Zimbabwe), ELICIT (Tanzania), and MAL-ED studies—that examined inflammation and growth factor pathways.

• The trial was conducted in a single urban Indian setting, which limits extrapolation to rural or diverse socioeconomic contexts. The discussion should acknowledge this limitation more explicitly and suggest strategies for replication in varied environments.

5. Policy and Program Implications

• The conclusion is based on the non-significant findings of biomarkers. Whereas the short duration of biomarker assessment may oversimplify complex biological processes. More elaborate discussion is needed on possible confounders like infections, duration, and type of breastfeeding.

Minor Comments

1. Abstract: Conclude with a stronger statement about contribution: e.g., “These findings add to the understanding of biological mechanisms underlying integrated early-life interventions in LMICs.”

2. Tables: Present only adjusted results in the main text; unadjusted data may be submitted as supplementary files. Ensure all tables include units (mg/L, ng/mL) and consistent decimal formatting.

3. CONSORT Diagram: Please include the number of exclusions, losses to follow-up, and reasons for non-participation in Figure 1 for transparency.

4. Discussion: Add a short note acknowledging that biomarker variability in early infancy is high and may obscure subtle intervention effects.

5. References: Consider citing more recent literature (published within the last 3 years) that links microbiome–inflammation–growth relationships in infants.

6. Language and Formatting: Ensure consistency in abbreviations (e.g., IGFBP3 vs IGF-BP3). Use consistent phrasing for “preconception, pregnancy, and early childhood interventions, growth-related biomarkers, and growth factor profiles” throughout.

Overall Recommendations: Minor–to–Moderate Revision

This is a robust, well-implemented study addressing an important mechanistic question within global child health. Although the results are null, they offer valuable insights into early-life biology and integrated program evaluation. Strengthening the biological framing, contextual discussion, and presentation of adjusted analyses will substantially enhance the manuscript’s impact and readability.

**Do you want your identity to be public for this peer review?** For information about this choice, including consent withdrawal, please see our Privacy Policy

Reviewer #1: No

Reviewer #2: **Yes: ** Dr. Quazi Syed Zahiruddin

Reviewer #3: **Yes: ** Abhay M Gaidhane

---

## [Editor Report · Decision Letter 1]

26 Nov 2025

Impact of an Integrated Intervention Package During Preconception, Pregnancy, and Early Childhood on inflammation, IGF-1, IGFBP3 during first 6 months of life: Findings from the WINGS Randomized Controlled Trial

PGPH-D-25-02277R1

Dear Dr. Chowdhury,

We are pleased to inform you that your manuscript 'Impact of an Integrated Intervention Package During Preconception, Pregnancy, and Early Childhood on inflammation, IGF-1, IGFBP3 during first 6 months of life: Findings from the WINGS Randomized Controlled Trial' has been provisionally accepted for publication in PLOS Global Public Health.

Best regards,

Rahul Gajbhiye, MBBS PhD

Academic Editor

Thank you for your thorough revisions.